# Simulation and Techno-Economical Evaluation of a Microalgal Biofertilizer Production Process

**DOI:** 10.3390/biology11091359

**Published:** 2022-09-16

**Authors:** Juan Miguel Romero-García, Cynthia Victoria González-López, Celeste Brindley, José María Fernández-Sevilla, Francisco Gabriel Acién-Fernández

**Affiliations:** 1Department of Chemical, Environmental and Materials Engineering, Centre for Advanced Studies in Earth Sciences, Energy and Environment (CEACTEMA), Universidad de Jaén, Campus Las Lagunillas, 23071 Jaén, Spain; 2Department of Chemical Engineering, University of Almeria, 04120 Almería, Spain

**Keywords:** microalgae, biofertilizer, amino acids, wastewater, simulation, circular bioeconomy, techno-economical evaluation

## Abstract

**Simple Summary:**

The world’s population is expected to increase to almost 10,000 million by 2025, thus requiring an increase in agricultural production to meet the demand for food. Hence, an increase in fertilizer production will be needed, but with more environmentally sustainable fertilizers than those currently used. Traditional nitrogenous fertilizers (TNFs, inorganic compounds, for example nitrates and ammonium) are currently the most consumed. Biofertilizers concentrated in amino acids (BCAs) are a more sustainable alternative to TNF and could reduce the demand for TNFs. BCAs are widely used in intensive agriculture as growth and fruit formation enhancers, as well as in situations of stress for the plant, helping it to recover its vigor. In addition, BCAs minimize or contribute to reducing the damage caused by pests and diseases, have an immediate action, giving a full utilization and, lastly and most importantly, they produce savings in the crop. The objective of this work is to propose a process for the production of biofertilizer concentrated in free amino acids from microalgal biomass produced in a wastewater treatment plant and to carry out techno-economic evaluation in such a way as to determine the viability of the proposal.

**Abstract:**

Due to population growth in the coming years, an increase in agricultural production will soon be mandatory, thus requiring fertilizers that are more environmentally sustainable than the currently most-consumed fertilizers since these are important contributors to climate change and water pollution. The objective of this work is the techno-economic evaluation of the production of biofertilizer concentrated in free amino acids from microalgal biomass produced in a wastewater treatment plant, to determine its economic viability. A process proposal has been made in six stages that have been modelled and simulated with the ASPEN Plus simulator. A profitability analysis has been carried out using a Box–Behnken-type response surface statistical design with three factors—the cost of the biomass sludge, the cost of the enzymes, and the sale price of the biofertilizer. It was found that the most influential factor in profitability is the sale price of the biofertilizer. According to a proposed representative base case, in which the cost of the biomass sludge is set to 0.5 EUR/kg, the cost of the enzymes to 20.0 EUR/kg, and the sale price of the biofertilizer to 3.5 EUR/kg, which are reasonable costs, it is concluded that the production of the biofertilizer would be economically viable.

## 1. Introduction

The increase in the world population, which by 2050 is expected to be close to 10,000 million [1,2], will require an increase in agricultural production to meet food needs, for which more fertilizers will be needed. These fertilizers should be more environmentally sustainable since traditional fertilizers are important contributors to climate change and water pollution [3,4]. In this context, biofertilizers and, specifically, free-amino-acid concentrates are presented as an alternative in the advancement towards a sustainable food system for the 21st century, in line with the Sustainable Development Goal 2 (SDG2) of the United Nations’ 2030 Agenda, zero hunger, but also connected to SDG6, clean water and sanitation, and SDG14, life below water, and in the circular bioeconomy framework [5,6].

Since the beginning of the 19th century, plant physiologists have shown that, in addition to carbon, hydrogen, and oxygen, thirteen chemical elements are considered essential for plant life, of which the most important, by far, is nitrogen. Global fertilizer consumption is in the order of 200 Mt with around 110 Mt corresponding to nitrogenous fertilizers [7]. The annual fertilizer consumption in Spain is in the order of 5 Mt, almost half being simple nitrogenous fertilizers [8]. Traditional fertilization (inorganic compounds, for example, nitrates and ammonium) does not always achieve its objective. Situations of hydric, thermal, or phytotoxic stress can prevent plants from absorbing available nitrogen and using it for their biosynthetic processes [9]. In addition, processes used to produce these traditional fertilizers are highly polluting, making them unsustainable over time and driving the change taking place in recent years towards organic farming, in which these inorganic fertilizers cannot be used but, instead, only organic fertilizers are allowed [10].

These problems can be solved by making use of the most modern knowledge of plant physiology, using basic elements of biosynthesis, that is, amino acids. Amino acids are organic compounds that constitute the fundamental basis of many biological molecules: they are fundamental structural units for the formation of proteins (complex macromolecules that develop structural functions as components of cell walls in plants), enzymes (biochemical catalysts in many processes), and the starting materials for the synthesis of other essential substances. No biological process can be carried out without amino acids taking part in some steps [11].

Plants synthesize amino acids through enzymatic reactions, via amination and transamination processes, which involve a large energy expenditure by the plant. Starting from the nitrogen cycle, the possibility of being able to supply amino acids to the plant is proposed, saving it from having to synthesize them, thus achieving a better and faster response. In this way, the amino acids are quickly used by the plants, and their transport takes place as soon as they are applied, being distributed throughout the plant, especially to the growing organs. Amino acids, in addition to having a nutritional function, can act as regulators of the transport of microelements, since they can form complexes with metals in the form of chelates [12].

The use of amino acids in crops favors seed germination, vegetative growth, flowering, fruit setting, fattening, and fruit ripening; it advances the recovery of crops that have been subjected to adverse circumstances (frost, hail, drought, phytotoxic effects of pesticides, etc.), improves fruit quality, increases production, and delays ageing. Applied via the roots, they favor the development of bacterial flora and beneficial microorganisms, thus increasing the availability of nutrients for crops. In addition, they minimize or contribute to reducing the damage caused by pests and diseases, have an immediate action and a full utilization and, lastly and most importantly, produce savings in the crop [13].

Amino acids are obtained by hydrolysis of proteins until a certain degree of hydrolysis is achieved. After the hydrolysis process, a mixture composed mainly of free amino acids is obtained, although, to a lesser extent, it also contains small chains of amino acids (short-chain peptides); this fraction is separated by centrifugation to increase the concentration of the organic fertilizer. The plant can only use free amino acids and, within these, only those that are in their L-enantiomeric forms.

The proteins used to obtain amino acids can be of plant or animal origin. The best ones for use as fertilizers are proteins of vegetable origin since they contain the amino acids that plants use and in the usual concentrations found in plants [14]. In this work, the biomass of microalgae obtained as a by-product of wastewater treatment is used as raw material, and its processing is directed to the production of biofertilizer concentrated in amino acids, with the consequent advantages to the environment that this approach entails. Among the species of microalgae that have been used most frequently for the production of biofertilizers and biostimulants are *Chlorella* sp., *Scenedesmus* sp., *Dunaliella* sp., *Nannochloropsis* sp., *Haematococcus* sp., and *Chlamydopodium* sp. [15,16,17,18].

The aims of this work are the design a production process for biofertilizer concentrated in free amino acids derived from microalgal biomass produced in a wastewater treatment plant and to carry out simulation with Aspen Plus and techno-economic evaluation of the design process. The biofertilizer produced will have a minimum content of free amino acids of 6% to comply with current legislation in Spain and Europe, in which the biostimulant capacity of amino acids has already been included (Royal Decree 506/28 June 2013, on fertilizer products (B.O.E.; 7 October 2013); Order APA/104/11 February 2022, which modifies annexes I, II, III and VI of the Royal Decree 506/28 June 2013, on fertilizer products; and Regulation (EU) 2019/1009 of the European Parliament and of the Council of 5 June 2019, which establishes provisions relating to the availability on the market of EU fertilizer products).

## 2. Materials and Methods

### 2.1. Raw Material

To develop this work, the starting point was an existing urban wastewater treatment plant. Unlike conventional processes, in this case, the treatment plant used microalgae. Once the biological treatment had been carried out and the regenerated water had been separated, microalgal biomass was obtained. This constitutes the raw material used in the proposed work. The selected microalgal species was *Scenedesmus almeriensis*, which has an average total protein content of 49.5% dry weight, estimated from total protein content data presented in several publications [19,20,21,22]. This biomass is considered suitable for obtaining amino acids due to its high protein content. Likewise, it has been confirmed that the *Scenedesmus almeriensis* hydrolysate has a good capacity as a biofertilizer and bio-stimulant [23,24].

To design the process, we considered that the microalgal biomass was produced in a 5 ha wastewater treatment plant, similar to that of AQUALIA S.L in Mérida, Spain [25]. In this plant, the open reactor installation yielded an average biomass productivity of 25 g/(m^2^·day), estimated from biomass productivity data presented in several publications [26,27,28,29,30], with an operating time of 330 days per year. According to these values, 82.5 t/(ha·year) of biomass would be produced. However, it must be taken into account that in these processes the biomass is harvested together with a considerable amount of water, since after the biological treatment it goes to a downstream stage where the reclaimed water is separated (using ultrafiltration–centrifugation membranes), finally obtaining a sludge that typically consists of 20% biomass and 80% water.

### 2.2. Proposed Process

The proposed process consists of a mechanical pressure pretreatment of the microalgal biomass (high-pressure homogenization) followed by enzymatic hydrolysis to obtain a hydrolysate concentrated in free L-amino acids with value as a biofertilizer and biostimulant. In the hydrolysis stage, the temperature is controlled by supplying heat from a solar collection system. The separation between the biomass residue and the free L-amino acid concentrate product is carried out by centrifugation. The free L-amino acids concentrate is subsequently stored until it is packaged in suitable containers for production of the final biofertilizer product ready for market release.

#### 2.2.1. Biomass Storage

The biomass used for amino acid production comes from a wastewater treatment plant, in which the nutrients present in the wastewater are used to produce reclaimed water and the biomass of interest, using microalgae for this conversion. The wastewater treatment plant produces around 2000 t/year of sludge from a centrifugation stage, with a biomass concentration of 200 kg/m^3^. This sludge has a pH of 8 and a density of 1.03 t/m^3^. The plant is in operation for 330 days a year, 24 h a day, which will be the same period in which the biofertilizer production plant will operate.

Biomass sludge storage time is set to one day (24 h) to avoid decomposition problems.

#### 2.2.2. Microalgal Biomass Pretreatment

A mechanical pretreatment is carried out using high-pressure homogenization. For this purpose, the microalgal sludge obtained from the wastewater treatment plant is subjected to a pressure of 200 bar followed by depressurization to ambient pressure. This allows a better degree of protein hydrolysis, in addition to conferring biostimulant properties to the hydrolysate produced [23,24]. According to Navarro-López et al. [24], the pressure treatment confers a character as a biostimulant with a capacity to improve the germination index by 10%.

#### 2.2.3. Protein Hydrolysis

Enzymatic hydrolysis is chosen because it is a highly selective process that has multiple advantages: it does not destroy amino acids; all amino acids are in their L form (natural form) usable by plants; no organic or amine nitrogen is formed; and a high percentage of biological and nutritional value is achieved. The bibliographical references show that the most convenient way to carry out the hydrolysis is to use two types of proteases, endoproteases and exoproteases, with one acting immediately after the other [20,22,23,24].

The hydrolysis process proposed by Romero-García et al. [21] is used, since they used *Scenedesmus almeriensis* biomass as a substrate with a concentration of 200 g/L, as in this work. The following commercial enzyme preparations from Novozymes A/S are used in the process: Alcalase 2.5 L. (endoprotease activity) and Flavourzyme 1000 L (exoprotease activity), which are in liquid form [31]. The hydrolysis time is 3 h and the operating temperature is 50 °C. To supply the necessary heat for the hydrolysis process, a low-temperature solar installation is used. By means of a heat exchanger, the solution to be hydrolyzed is heated using the hot fluid from the solar collection system. The dose of enzymes will be 4% (*v*/*w*) to the substrate (microalgal biomass). Alcalase 2.5 L is first added to the substrate at pH = 8.0. After 2 h, the pH is adjusted to a value of 7.0 and Flavorzyme 1000 L is added. The degree of protein hydrolysis reached after 3 h is 55% [21].

In the process, Ca(OH)_2_ at 70% (*w*/*v*) is used, obtaining the Ca(OH)_2_/biomass sludge ratio experimentally, resulting in a value of 0.79% (*v*/*v*). Ca(OH)_2_ is added over 2 h, during which the pH changes due to the action of Alcalase 2.5 L. To adjust the optimal pH of Flavourzyme 1000 L, 98% wt. sulfuric acid (H_2_SO_4_) is added, obtaining an experimental sulfuric acid/biomass sludge ratio with a value of 0.19% (*v*/*v*). All of the sulfuric acid added reacts to form gypsum (CaSO_4_). Ca(OH)_2_ (70% (*w*/*v*)) and H_2_SO_4_ (98% wt.) are stored in sufficient quantity for one month of operation. The necessary heat to reach and maintain the optimum hydrolysis temperature of 50 °C is provided by the solar-heat-capture system.

#### 2.2.4. Centrifugation

The sedimentation alternative was ruled out due to the small particle diameter of the biomass and the low difference in density between the fluid and the solid. The main factor affecting the economics of centrifuge operation is particle size. In filtration, the choice of filter media depends on particle size, but overall economics are not affected. The cut-off point according to the cost that determines the choice of separation by ultrafiltration and by centrifugation corresponds to the interval of 1–2 μm, the size of *Escherichia coli* [32]. Since the cells of microalgae are about ten times larger than those of bacteria, filtration as a separation operation was ruled out and centrifugation was selected.

During this stage, the stream exiting the hydrolysis reactor is separated in two parts, one containing the free-amino-acid concentrate (product) and the other containing the biomass remains (by-product, 20% of the input volume to the centrifuge). In the by-product stream, it is assumed that 100% of the solids remains are retained, which is generally a good approximation as centrifugation is a highly efficient operation.

#### 2.2.5. Free-Amino-Acid Concentrate Storage and Packaging

Once the free-amino-acid concentrate is obtained, it is stored in a tank with a maximum capacity that is equivalent to 30 days of production. Next, the free-amino acid concentrate is packaged in containers of different volumes to obtain the final biofertilizer market product. The volumes of the containers are 1, 5, 10, and 20 liters, which are the most commonly used in the agricultural-fertilizer market.

#### 2.2.6. Solar Thermal Collector

To supply the necessary heat for the hydrolysis process, a low-temperature solar installation is used. To heat the process water (the heating fluid in the heat exchanger), the heat accumulation system used consists of a collection system of thermal collectors and a tank storage system.

##### Heat-Storage System

To design the tank of this system an autonomous 3-day operation was considered, with which, having a large volume, greater thermal inertia was achieved.

##### Heat-Capture System

Solar energy is captured utilizing thermal collectors, so capture area depends on the energy required. Considering that there is a 20% energy loss, the collection system is designed to collect 20% more heat than that consumed during one day of operation, thus ensuring the necessary heat collection. A collection time of 8 h and a minimum temperature variation in the collectors of 10 °C were considered.

### 2.3. Process Modelling and Simulation: Material and Energy Balances

For the modelling and simulation of the stages of the proposed process, and therefore for obtaining the material and energy balances, the modular simulator ASPEN Plus V9 (Aspen Technology Inc., Bedford, MA 01730 EE.UU., USA was used. A processing capacity of 2062.5 t/year of biomass sludge with 7920 h/year of operation was considered. The thermodynamic models used were non-random two liquids (NRTL) and the Hayden–O’Connell equation of state (HOC EoS). Figure 1 shows the process flow diagram in ASPEN Plus. Table 1 describes the most significant aspects in the modelling of each piece of the equipment.

### 2.4. Economic Analysis

This stage consists of two parts: (1) Selection and design of each equipment, estimation of their costs, and, subsequently, estimation of total investment capital (CAPEX) and (2) determination of the total annual cost of operation (OPEX), as well as the expected annual income. With all the above, economic analysis of the proposed process was carried out to determine its profitability. This was performed using the Aspen Process Economic Analyzer V9 software (APEA, Aspen Technology Inc., Bedford, MA 01730 EE.UU., USA) and the mass and energy balances from the process simulation carried out previously. As input data, a project life of 10 years (7920 h/year) has been considered, which is a common value for biotechnological projects [33,34]; a conservative annual interest rate of 5% with the current value that could be around 2.5%; taxes on profits of 25% (current value in Spain); and the straight-line method for calculating depreciation. Regarding the costs related to labor, costs associated with operators and supervisors of 23.41 EUR/h and 26.45 EUR/h, respectively [35], have been considered. As for the cost of the Ca(OH)_2_ and sulfuric acid, 65 EUR/t and 73 EUR/t, respectively, have been considered. The price of electricity has been set at 0.12 EUR/kWh (average price for 2021 in Spain) and the price of water at 1 EUR/m^3^.

The cost of biomass sludge and enzymes is variable, so different scenarios have been considered when carrying out a sensitivity analysis. The same has been done with the sale price of the biofertilizer. Thus, a sensitivity analysis has been carried out using a Box–Behnken-type response-surface statistical design with 13 scenarios (Table 2), in which three factors varied—the cost of the biomass sludge, the cost of the enzymes, and the sale price of the biofertilizer—determining the indicators of economic profitability (net present value (NPV), payback period (PP), internal rate of return (IRR), and profitability index (PI)).The cost of microalgal biomass in open photobioreactors ranges between 1 EUR/kg and 5 EUR/kg, according to the literature [27,30,33,36]. Therefore, if the sludge contains 20% biomass it should have a cost between 0.2 EUR/kg and 1 EUR/kg. In the case of the enzymes (Alcalase 2.5 L and Flavourzyme 1000 L), their cost would range between 10 EUR/kg and 25 EUR/kg [37,38,39]. And, finally, the market price for biofertilizers similar to the one produced in this process ranges between 2.5 EUR/kg and 7.5 EUR/kg, according to different references and websites consulted [40,41,42,43,44,45,46,47]. The results obtained were adjusted according to a quadratic model to be able to appreciate the influence of each of the factors studied in the answers (Y), according to Equation (1), using the Design-Expert 8.0.7.1 program (Stat-Ease Inc., Minneapolis, MN, USA):*Y* = *a*_0_ + *a*_1_A + *a*_2_B + *a*_3_C+ *a*_4_AB + *a*_5_AC + *a*_6_CB + *a*_7_A^2^+ *a*_8_B^2^ + *a*_9_C^2^(1)
where A is the cost of the biomass sludge (EUR/kg), B is the cost of the enzymes (EUR/kg), and C is the sale price of the biofertilizer (EUR/kg).

The model enables the influence of each factor on the responses to be determined as well as the interactions between factors, according to the *a_i_* coefficients. A higher absolute value of the coefficient in terms of coded values implies a larger effect on the factor. The sign indicates if the answer is positive or negative. *a*_0_ indicates the response values at the center point (coded value = 0). There are three coefficients for factors (*a*_1_–*a*_3_). The fit is hierarchical, there will only be interactions and quadratic coefficients if the factor coefficient (*a*_4_–*a*_9_) exists.

## 3. Results and Discussion

### 3.1. Production of Biofertilizer and Consumption of Reagents and Heating Energy per Year

Table 3 shows the main characteristics of the process streams and the results obtained from the resolution of the materials and energy balances with Aspen Plus.

Once all the streams that appear in the process had been characterized, the needs for reagents and energy, and the production of biofertilizer for one year were calculated. Table 4 shows the values obtained, resulting in a biofertilizer production of 1645.95 t. A solid by-product was also produced and this could be used as an organic amendment with an amount of 483.52 t.

### 3.2. Equipment Selection, Sizing, and Cost Estimation

Except for the packaging machine and the solar thermal collectors, the rest of the equipment has been selected, dimensioned, and its cost estimated using Aspen Process Economic Analyzer V9 (APEA) and its databases, all following the materials and energy balances and the process design considerations. Table 5 shows the selected equipment, the main dimensions, the number of pieces of equipment, and the cost. The total cost of equipment is EUR 1,088,448.80.

#### 3.2.1. Liquid Packaging Machine

A Flowmatic™-Liquid Mass model packaging machine from Capmatic Ltd. [48] was selected since it meets the specifications. It allows containers of very different volumes up to 20 L to be filled, with a packaging capacity of up to 120 bottles per minute. It is also specially designed for viscous and semi-viscous liquids and is made of 316 L stainless steel. It is budgeted at EUR 80,000, with Euroguard (CE) system and transport costs.

#### 3.2.2. Thermal Solar Collectors

ISONOX II brand collectors (Isofoton, S.A, Malaga, Spain [49]) are used, with the following characteristics: surface per collector (m^2^) = 1.9; collection capacity (kJ/h m^2^·°C) = 350.22; absorptivity = 0.95; emissivity = 0.05; maximum temperature (°C) = 180. The cost per thermal collector is EUR 650.

Considering the value of heat that the collection system has to supply to the process (10,336.50 kJ/h, Table 3) and the considerations made in Section 2.2.6 (20% losses, 8 h of collection, and 10 °C of minimum temperature variation in the collectors) and the specifications of the selected collector, a heat capture surface of 10.63 m^2^ is required, which is equivalent to 5.6 collectors, so a total of seven collectors have to be installed.

### 3.3. Investment Capital

Once the total cost of equipment was estimated using APEA software, the initial investment capital required (CAPEX) was estimated, resulting in a value of EUR 9,648,523.33 for this project. The item with the greatest weight was “design, engineering and acquisitions” (Table 6). This investment value is in the order of other plants with very small treatment capacities of around 2000 t/year, with investment values of less than EUR 9 million [50,51].

### 3.4. Economic Sensitivity Analysis

Table 7 shows the results of the statistical design carried out for the sensitivity analysis. As can be seen that in conditions 1, 9, and 11, the NPV is negative, that is, there will be no benefit in the life of the project and, therefore, no value can be obtained from the PP since the investment will never be recovered nor the IRR value since it cannot be less than 0; and in terms of PI, negative values are obtained, which indicate how much would be lost for each euro of investment if these conditions were met, ranging from −0.08 to −1.02.

The results obtained have been modelled and the values of the coefficients of the models and the response surfaces obtained for the four responses studied are shown in Figure 2, Figure 3, Figure 4 and Figure 5. Of the three factors studied, the most influential, by far, is the sale price of the biofertilizer, followed by the cost of the biomass sludge and, finally, the cost of the enzyme. If, for example, the coded coefficients obtained for the NPV model are compared, the sale price of the biofertilizer is almost 5 times higher than the cost of the biomass sludge and 14 times the cost of the enzyme. Similar results were obtained when a sensitivity study about the production of microalga protein concentrate by flash hydrolysis was performed, showing that the sale price of the concentrate had the greatest influence [52]. Thus, the sale price of the final product is crucial for the economic viability of microalgae valorization processes, as shown in a multi-objective study of techno-economic optimization of the microalgae-based value chain; the results obtained here are in correlation with those found in the literature [53].

If the response surfaces for the NPV (Figure 2) and the PI (Figure 5) are observed, a flat zone below the value 0 of each of these variables indicates that there would be no benefit and the project would not make a profit. In the case of the higher costs of biomass sludge and enzymes, a sale price of the biofertilizer higher than 3.5 EUR/kg would be needed to achieve NPV > 0 and IR > 0. If NPV = 0 and IR = 0, it means that the initial investment is recovered in the life of the project, but it would not be an attractive project for investors and, therefore, it would not be carried out.

For projects to be attractive, the recovery of the investment must occur as soon as possible, with around 60–70% of the life of the project being acceptable. In the present case, the PP should lie between 6 and 7 years. If higher costs of biomass sludge and enzymes are considered to achieve a TRI = 6–7 years, the sale price of the biofertilizer should be between 3.9–4.4 EUR/kg. And, finally, the IRR must be as high as possible, but it must at least triple the interest rate considered, which in this case is 5%, therefore the IRR should be greater than 15%. To achieve this IRR > 15% with the higher costs of biomass sludge and enzymes, the sale price of the biofertilizer would have to be greater than 4.1 EUR/kg.

### 3.5. Case Study

Once the sensitivity analysis had been carried out, the case study was proposed, in which specific values are given to the three factors studied. In the case of the cost of biomass sludge, a value of 0.5 EUR/kg is used, which corresponds to a biomass cost of 2.5 EUR/kg, being conservative regarding biomass costs from wastewater residuals in open reactors, which are often below 2 EUR/kg [36]. The cost of the enzymes is established at 20 EUR/kg, which is acceptable according to the values found in the literature, and finally, a value of 3.5 EUR/kg for the sale of the biofertilizer, which is below the reference price for this product, so as to facilitate its market entry. The price of the biofertilizer has also been chosen to be conservative and to take into account the effect that a lower efficiency of the enzymes and/or a lower concentration of the microalgal sludge could have on the system. It is clear that if the concentration of the sludge is maintained and the efficiency of the enzymes is lower, a lower concentration of amino acids would be produced. In the same way, if the concentration of biomass sludge is lower and the efficiency of the enzymes is maintained, the concentration of free amino acids would also be lower.

With the values proposed above an annual production cost of EUR 4.22 million was obtained in the case study, with the cost of raw materials being the factor with the greatest weight at over 40% of the total (see Table 8). The NPV at the end of the 10-year life of the project is EUR 9.17 million and its evolution throughout the project can be seen in Figure 6. The IR obtained is 0.95, that is, a profit of almost one euro per euro invested will be achieved, which means that after 10 years the profit will have almost doubled the amount that was invested, around 10% per year, much higher than the current interest rates on bank deposits, which are below 0.1%. In addition, the PP is 6.5 years and the IRR > 18%, a value that triples the 5% interest considered for the project. The values obtained here are similar to those of another very small plant that uses around 1000 t/year of grape pomace, in which a PP = 5.8 years, IRR = 13.2%, and NPV = €3.58 million were obtained [51].

It is worth mentioning that in the economic analysis carried out, the income that would be obtained from the sale of the solid by-product has not been taken into account. If it were sold as an organic amendment at a price of around 60 EUR/t [54], it would result in an annual amount of around EUR 29,000, which means that in over ten years of life it would be close to EUR 300,000.

On the other hand, the sustainability of the proposed process should be highlighted, since a previous study by Arashiro et al. [55] carried out a life-cycle analysis in which two scenarios for the environmental impact of the final use of the microalgal biomass produced in the treatment of wastewater were considered: (1) its use in an anaerobic digestion process to produce biogas and (2) its use for the production of biofertilizer. It turned out that the second scenario was more environmentally friendly in 7 of 11 impact categories (climate change, ozone depletion, freshwater eutrophication, marine eutrophication, photochemical oxidant formation, fossil depletion, and human toxicity). In that same study, these two scenarios were compared with the traditional treatment of wastewater with activated sludge, resulting in a lower impact in 6 of 11 impact categories (climate change, ozone depletion, freshwater eutrophication, marine eutrophication, photochemical oxidant formation, and fossil depletion) [55].

The company Biorizon Biotech SL, a world pioneer in the development of various biofertilizers and biostimulants based on microalgae, has taken an important step towards strengthening corporate sustainability by joining the Global Compact for United Nations because it aims to promote sustainable economic development and contribute to minimizing the nitrogen footprint [56]. Other companies such as Algaenergy SA, also produce biofertilizers based on microalgae, stating that they provide three types of general benefits to crops: higher yield, better quality, and greater stress resistance. This company also affirms that its products contribute decisively to the conservation of nature and the environment since, for every 5 L produced, 2 kg of CO_2_ is removed during the biomass production process [57].

On the other hand, to advance and promote compliance with the Sustainable Development Goals of the United Nations’ 2030 Agenda, this agency organizes the program for Progress and Development, in which the most outstanding projects are selected. Within the Ocean Innovation Challenge (OIC), a Spanish company, Ficosterra, dedicated to the transformation of algae into fertilizers, biostimulants, and biofertilizers for agriculture, has been chosen from among over 600 candidates. With their project “Nutrialgae” they propose to demonstrate that the use of biostimulants reduces water contamination while increasing the productivity of crops by up to 15%, on average, thus advancing towards the agriculture of the 21st century [58,59].

## 4. Conclusions

A process has been proposed for the production of a biofertilizer concentrated in free L-amino acids from microalgal biomass, which allows a concentration of 6% of free amino acids to be obtained, thus complying with current legislation on fertilizers in Spain and the European Union. The process transforms the microalgal biomass produced in a wastewater treatment plant into a biofertilizer that can be used in agriculture to improve crops and reduce the use of traditional fertilizers, thereby advancing towards a sustainable food system for the 21st century, in line with the Sustainable Development Goals of the United Nations’ 2030 Agenda.

The ASPEN Plus simulator allowed the process to be modelled and the material and energy balances necessary for the following stages of analysis to be obtained. The Aspen Process Economic Analyzer software has provided estimates of the CAPEX and OPEX, finding that the greatest weight is for the cost of raw materials.

The sensitivity analysis carried out using a Box–Behnken-type response surface statistical design with three factors—the cost of the biomass sludge, the cost of the enzymes, and the sale price of the biofertilizer—determined that the most influential factor in profitability is the sale price of the biofertilizer. In the base case proposed, the cost of the biomass sludge (0.5 EUR/kg) and enzymes (20.0 EUR/kg) and the sale price of the biofertilizer (3.5 EUR/kg) have been set, showing that the production of the biofertilizer is economically feasible, with an NPV = EUR9.17 million, IR = 0.95, PP = 6.5 years, and IRR = 18.31%, for a project life of 10 years.

## Figures and Tables

**Figure 1 biology-11-01359-f001:**
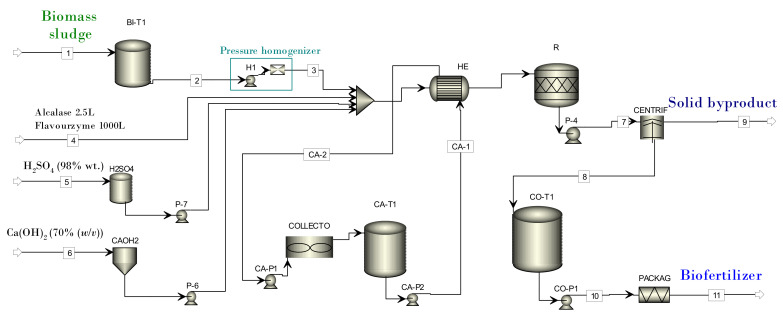
The process flow diagram in ASPEN Plus.

**Figure 2 biology-11-01359-f002:**
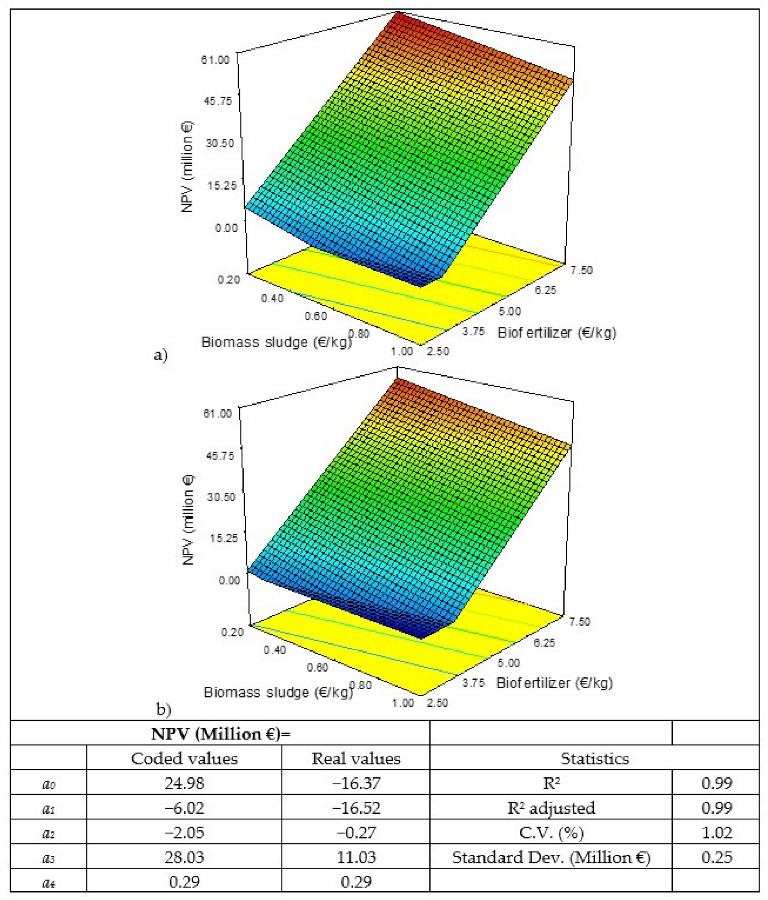
Results of the statistical analysis of the net present value (NPV): response surface, model in coded and real values, and statistics. Enzyme cost: (**a**) 10 EUR/kg and (**b**) 25 EUR/kg.

**Figure 3 biology-11-01359-f003:**
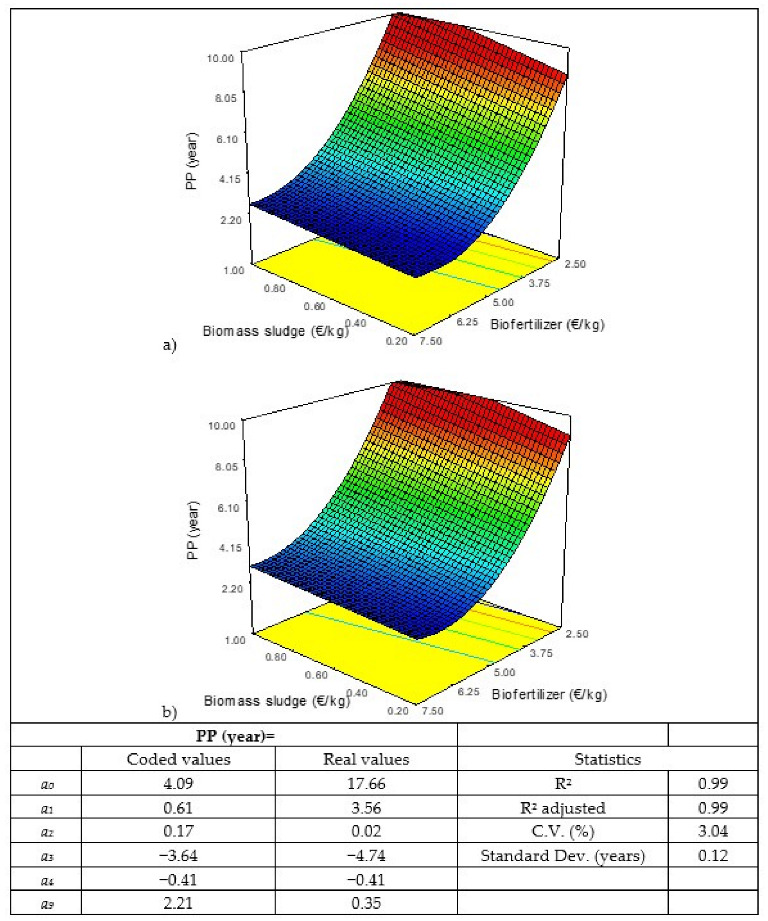
Results of the statistical analysis of the payback period (PP): response surface, model in coded and real values, and statistics. Enzyme cost: (**a**) 10 EUR/kg and (**b**) 25 EUR/kg.

**Figure 4 biology-11-01359-f004:**
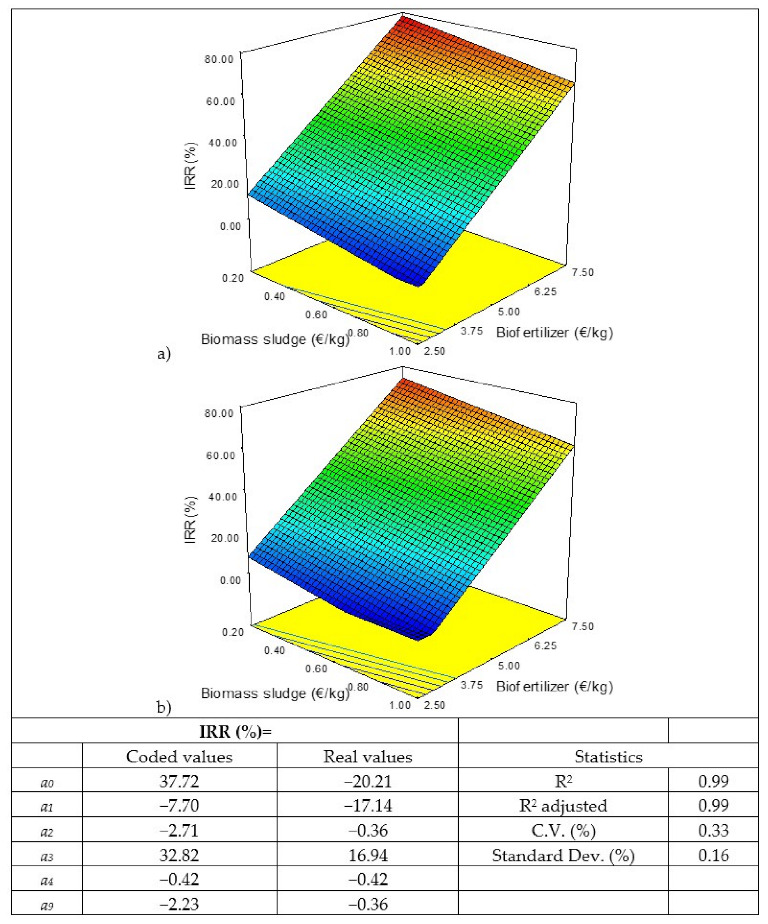
Results of the statistical analysis of the internal rate of return (IRR): response surface, model in coded and real values, and statistics. Enzyme cost: (**a**) 10 EUR/kg and (**b**) 25 EUR/kg.

**Figure 5 biology-11-01359-f005:**
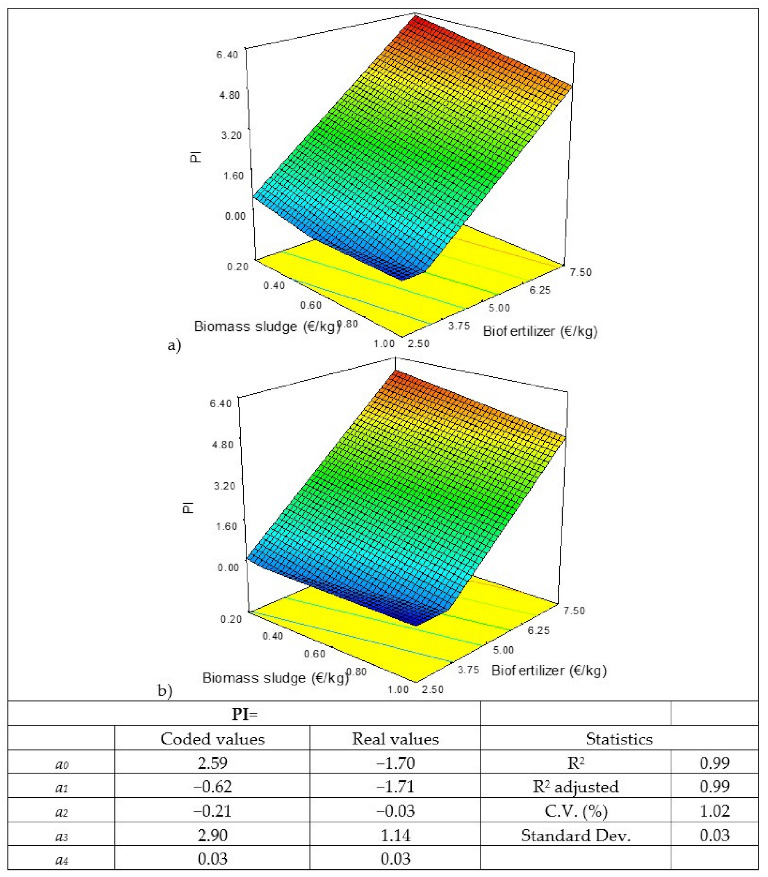
Results of the statistical analysis of the profitability index (PI): response surface, model in coded and real values, and statistics. Enzyme cost: (**a**) 10 EUR/kg and (**b**) 25 EUR/kg.

**Figure 6 biology-11-01359-f006:**
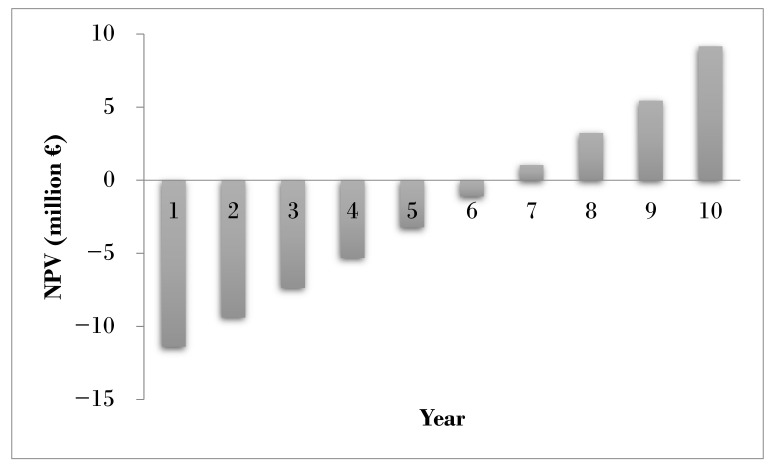
Evolution of the net present value (NPV) during the years of life of the project.

**Table 1 biology-11-01359-t001:** Modelling of equipment in ASPEN Plus.

Process Stage	Equipment Name	Modelling
Biomass storage (BIOMA-ST)	BI-T1	Tank
Pretreatment (PRETRE)	H1	High-pressure homogenizer: a pump to increase the pressure up to 200 bar and a valve to depressurize
Enzymatic hydrolysis (HYDROLY)	P-6	Pump
	R	Hydrolysis reactor: stoichiometric reactor in which two reactions take place: first, the hydrolysis of proteins with a conversion of 55%, and second, the reaction of H_2_SO_4_ to completion
	P-7	Pump
	P-4	Pump
	HE	Heat exchanger: countercurrent, with a hot fluid outlet temperature of 65 °C and a minimum approach temperature of 10 °C
	CaOH_2_	Tank
	H_2_SO_4_	Tank
Centrifugation (CENTRIFU)	CENTRIF	Centrifuge: disc centrifuge to achieve 35% humidity in the solid
Concentrate storage (CONCE-ST)	CO-T1	Tank
	CO-P1	Pump
Packaging (PACKAG)	PACKAG	Liquid packaging machine: no model in ASPEN Plus
Heat capture (HEAT-CAP)	CA-T1	Tank
	CA-P1	Pump
	CA-P2	Pump
	COLLECTO	Solar thermal collector: a heater to reach 85 °C, which allows the energy that the collectors will need to capture to be determined

**Table 2 biology-11-01359-t002:** Statistical design for sensitivity analysis.

Scenarios	Cost of the Biomass Sludge (EUR/kg)	Cost of the Enzymes (EUR/kg)	Biofertilizer Sale Price (EUR/kg)
1	1.0	17.5	2.5
2	1.0	25.0	5.0
3	0.6	10.0	7.5
4	0.2	25.0	5.0
5	0.6	17.5	5.0
6	0.2	10.0	5.0
7	0.2	17.5	2.5
8	1.0	10.0	5.0
9	0.6	25.0	2.5
10	1.0	17.5	7.5
11	0.6	10.0	2.5
12	0.2	17.5	7.5
13	0.6	25.0	7.5

**Table 3 biology-11-01359-t003:** Temperature, pressure, and flows of the main streams of the process.

Stream	1	2	3	4	5	6	7	8	9	10	11	CA-1	CA-2
T (°C)	25	25	40.04	25	25	25	50	50	50	50	50	85	65
P (bar)	1.01	1.01	1.01	1.01	1.01	1.01	1.01	1.01	1.01	1.01	1.01	1.01	1.01
Total flow (kg/h)	260.42	260.42	260.42	4.79	0.88	2.79	268.87	207.82	61.05	207.82	207.82	118.20	118.20
Water (kg/h)	209.85	209.85	209.85	2.4	0.02	1.39	213.98	194.03	19.95	194.03	194.03	118.20	118.20
Protein (kg/h)	25.03	25.03	25.03				11.26		11.26				
Lipids (kg/h)	8.85	8.85	8.85				8.85		8.85				
Carbohydrates (kg/h)	13.15	13.15	13.15				13.15		13.15				
Ash (kg/h)	3.54	3.54	3.54				3.54		3.54				
Enzymes (kg/h)				0.94			0.94		0.94				
Sucrose (kg/h)				1.44			1.44	1.31	0.13	1.31	1.31		
H_2_SO_4_ (kg/h)					0.86								
Ca(OH)_2_ (kg/h)						1.40	0.75		0.75				
Amino acids (kg/h)							13.77	12.48	1.28	12.48	12.48		
CaSO_4_ (kg/h)							1.2		1.2				
Supplied heat (kJ/h)												10,336.50	

**Table 4 biology-11-01359-t004:** Annual consumption of reagents and heating energy and annual production of solid by-products and biofertilizer.

Annual Values of Consumption and Products	Amount
Biomass sludge consumption (20%) (t)	2062.50
Enzyme consumption (t)	37.91
CaOH_2_ (70% *w*/*v*) consumption (t)	22.08
H_2_SO_4_ (98% wt.) consumption (t)	6.97
Energy consumption for heating (kJ)	8.19·10^7^
Production of solid by-product (t)	483.51
Biofertilizer production (t)	1645.95

**Table 5 biology-11-01359-t005:** Selected equipment, main parameters, and estimated cost.

Equipment Name	Selected Equipment	Units	Sizing		Cost (EUR)	Total Cost (EUR)	Source
BI-T1	Carbon steel storage tank	1	8.06	m^3^	24,300	24,300	APEA database
H1	SS316 stainless steel high-pressure positive displacement Pump	1	106	kW	229,100	229,100	APEA database
P-6	SS304 stainless steel centrifugal pump	1	1	kW	5700	5700	APEA database
R	SS304 stainless steel jacketed stirred tank	1	1.11	m^3^	78,200	78,200	APEA database
P-7	SS304 stainless steel centrifugal pump	1	1	kW	5700	5700	APEA database
P-4	Carbon steel centrifugal pump	1	1	kW	4500	4500	APEA database
HE	Carbon steel counterflow shell and tube heat exchanger	1	0.47	m^2^	8200	8200	APEA database
CaOH_2_	Fiber-reinforced polymer storage tank	1	3.8	m^3^	43,100	43,100	APEA database
H_2_SO_4_	Fiber-reinforced polymer storage tank	1	3.8	m^3^	43,100	43,100	APEA database
CENTRIF	Carbon steel high-speed disc centrifuge	1	254	mm	219,100	219,100	APEA database
CO-T1	Carbon steel storage tank	5	39.6	m^3^	60,000	300,000	APEA database
CO-P1	Carbon steel centrifugal pump	1	1	kW	4500	4500	APEA database
PACKAG	Liquid packaging machine Flowmatic™-Liquid Mass (Capmatic Ltd., Montreal, QC, Canada) for containers up to 20 L	1	120	bpm	80,000	80,000	Budget Capmatic Ltd., Canada
CA-T1	Carbon steel storage tank	1	11.4	m^3^	26,900	26,900	APEA database
CA-P1	Carbon steel centrifugal pump	1	1	kW	4500	4500	APEA database
CA-P2	Carbon steel centrifugal pump	1	1	kW	4500	4500	APEA database
COLLECTO	Flat solar collector ISONOX II (Isofotón, S.A, Malaga, SpainEspaña)	7	1.9	m^2^	650	4550	Budget Isofotón, S.A, Malaga, Spain

**Table 6 biology-11-01359-t006:** Investment capital for the project.

Item	Amount (EUR)	%
Equipment	1,088,448.80	11.28
Piping	542,566.50	5.62
Civil	405,450.50	4.20
Instrumentation	1,035,617.10	10.73
Electrical	1,301,905.00	13.49
Design, engineering, procurement	3,208,042.43	33.25
Administrative/contract taxes	576,675.40	5.98
Contingencies	1,489,817.60	15.44
**Total Investment**	9,648,523.33	100.00

**Table 7 biology-11-01359-t007:** Results of the statistical design of the sensitivity analysis. Net present value (NPV), payback period (PP), internal rate of return (IRR), and profitability index (PI).

Scenario	NPV (million EUR)	PP (Year)	IRR (%)	PI
1	−9.82			−1.02
2	17.19	5.02	27.46	1.78
3	54.97	2.58	71.01	5.70
4	28.94	3.65	42.70	3.00
5	25.09	4.01	37.62	2.60
6	32.99	3.35	48.18	3.42
7	3.11	8.92	9.96	0.32
8	21.24	4.44	32.64	2.20
9	−4.99			−0.52
10	47.07	2.88	60.29	4.88
11	−0.74			−0.08
12	58.82	2.46	76.53	6.10
13	50.92	2.72	65.42	5.28

**Table 8 biology-11-01359-t008:** Production cost per year and profitability parameters obtained for the case study.

Item	Amount (Million EUR/Year)	%
Raw-material cost	1.79	42.43
Utilities cost	0.30	7.11
Labor cost	0.58	13.74
Maintenance costs	0.06	1.38
Operating costs	0.15	3.44
Contingencies	0.32	7.56
Administration costs	0.26	6.05
Depreciation	0.77	18.28
**Production costs**	4.22	100.00
NPV (Million EUR)	9.17	
IRR (%)	18.31	
PP (year)	6.51	
IR	0.95	

## Data Availability

Not applicable.

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
