# Peer review of "Simulation and Techno-Economical Evaluation of a Microalgal Biofertilizer Production Process"

_biology, 2022, doi:10.3390/biology11091359_

Round 1
Reviewer 1 Report
The manuscript focuses on the production of biofertilizers. A wide range of economic analyses is carried out based on process modelling and simulation. In my opinion, the manuscript is interesting enough to be published, and its content adheres to the Biology journal standards.
I have several remarks which need the authors' attention (please check the attached file).

Author Response
- Line 129, 136, 278, 282 “Quoting multiple papers without providing details makes it difficult to understand the context. Give more details please.”
Answer: Thanks. The text has been modified to be clearer.
- Line 155 “Figures 2 and 3 show the same information. Please remove Fig.2 (block diagram) while leaving Figure 3 (flow diagram).”
Answer: Thanks. The figure 2 has been removed.
- Line 156 “The entire table should be located on one page. Check all the tables in the manuscript please.”
Answer: Thanks. Table 1 has been removed. All the tables have been checked to make sure that they appear in one page, if possible.
- Line 189 “Space between a value and a sign of the degree to remove. Check carefully all the manuscript please.”
Answer: Thanks for pointing out this observation. However, the International Bureau of Weights and Measures recommends leaving a space after the number when writing about temperature, whereas the University of Chicago Press and the Oxford University Press both employ no space between the number, the symbol, and the Latin letters representing Celsius or Fahrenheit, respectively. Therefore, since both are in common use, we have chosen the SI style.
- Line 194 “Use subscript, please. Check carefully all chemical formulas in the manuscript.”
Answer: Thanks. The symbol “2” in Ca(OH)2 is in fact presented in subscript, but unfortunately the style of text used does not actually lower the subscript in a very obvious way. All the chemical formulas have been checked.
- Line 197 “to remove”
Answer: Thanks. The word “to” has been removed
Reviewer 2 Report
Manuscript has been well written and the scientific data seems to be promising in terms of production of cost-effective environmental and sustainable microalgal biofertilizers with emphasis on techno-economic analysis is highly appreciated.
Author Response
Answer: Thanks.
Reviewer 3 Report
Manuscript No. Biology-1865574
Manuscript Title: Simulation of a process for the production of microalgal biofertilizer including techno-economical evaluation
Reviewer Comments
The manuscript is very interesting describing the economic feasibility aspects of microalgae based biofertilizers.
However, the following suggestions must be implemented before the manuscript is accepted for publication.
The title can be made more precise to describe the novelty highlights of the work. Words like “including” could be omitted in the title.
It is advisable to write the methodology in past tense.
3. Fig. 1 in Introduction could be removed as it is a common process flow chart for nitrogen fertilizer protein cycle. It is usually not advisable to keep such figures in the introduction section of a research paper.
4. Introduction section lacks paper related to the biofertilizer or biostimulatory potential of algal strains. Authors are advised to refer and cite more papers on this like doi.org/10.1016/j.indcrop.2020.112453 and doi.org/10.1016/j.biortech.2021.125588
5. Also, the discussion section of the manuscript needs to be strengthened. The study carried out must be compared with similar kind of TEA studies existing in literature to correlate with the similarity in sensitivity of parameters obtained.
6. Authors must discuss how the enzyme efficiency, variation in algal biomass concentration are going to affect the overall process feasibility of biofertilizer production and thereby its economics.
7. General comments: The entire manuscript must be thoroughly read and checked for any grammatical or typographical errors.

Author Response
- The title can be made more precise to describe the novelty highlights of the work. Words like “including” could be omitted in the title.
Answer: Thanks for making this suggestion. Title has been changed.
- It is advisable to write the methodology in past tense.
Answer: Thanks. The methodology has been re-written.
- Fig. 1in Introduction could be removed as it is a common process flow chart for nitrogen fertilizer protein cycle. It is usually not advisable to keep such figures in the introduction section of a research paper.
Answer: Thanks. Figure 1 has been removed.
- Introduction section lacks paper related to the biofertilizer or biostimulatory potential of algal strains. Authors are advised to refer and cite more papers on this like doi.org/10.1016/j.indcrop.2020.112453 and doi.org/10.1016/j.biortech.2021.125588
Answer: Thanks. References of articles related to the biofertilizer or the biostimulatory potential of microalgae strains have been added.
- Also, the discussion section of the manuscript needs to be strengthened. The study carried out must be compared with similar kind of TEA studies existing in literature to correlate with the similarity in sensitivity of parameters obtained.
Answer: Thanks for these suggestions. References and text have been added to the discussion.
- Authors must discuss how the enzyme efficiency, variation in algal biomass concentration are going to affect the overall process feasibility of biofertilizer production and thereby its economics.
Answer: Thanks. The effect on the viability of the process produced by a lower efficiency of the enzymes and a lower concentration of the microalgae sludge has been considered when selecting the sale value of the biofertilizer, selecting a conservative value, since we believe that it could be even higher if all the conditions of the proposed process were achieved. The discussion has been improved.
- General comments:The entire manuscript must be thoroughly read and checked for any grammatical or typographical errors.
Answer: Thanks. The manuscript has been revised and corrected.
Reviewer 4 Report
The article has publication potential since presents scientific interest and presents attractive information from a technical and economic point of view. But, the article needs some improvements presented below.
1) The Simple Summary not is necessary;
2) Avoid using terms like “our”;
3) Consider replacing the terms “milk of lime / lime milk” by “sodium hydroxide”;
4) Authors should clarify how the solution to be hydrolyzed is heated. Is the solution heated in the solar heating system or is it heated through a heat exchanger? Only by Figure 3 can it be understood. Also clarify which heating fluid is used in the heat exchanger;
5) Figure 2 and Table 1 are not necessary. Figure 3 and Table 2 are sufficient to understand the proposed system;
6) The authors report that the microalgae sludge must be pressurized to 200 bar but in Table 4 the pressure is constant in all streams. Apparently, stream 3 should have a higher pressure;
7) The authors clarify how they obtained the equipment costs but do not present information on how they estimated the other costs (Table 7);
8) In economic analysis results, compare the optimal sale price obtained for the biofertilizer (3.5 €/kg) with the standard selling price of a chemical fertilizer.
Author Response
1) The Simple Summary not is necessary;
Answer: Thanks, but the Simple Summary is mandatory in this journal.
2) Avoid using terms like “our”;
Answer: Thanks. The manuscript has been revised and corrected.
3) Consider replacing the terms “milk of lime / lime milk” by “sodium hydroxide”;
Answer: Thanks. Milk of lime/lime milk has been replaced by Ca(OH)2
4) Authors should clarify how the solution to be hydrolyzed is heated. Is the solution heated in the solar heating system or is it heated through a heat exchanger? Only by Figure 3 can it be understood. Also clarify which heating fluid is used in the heat exchanger;
Answer: Thanks. The text has been modified to be clearer.
5) Figure 2 and Table 1 are not necessary. Figure 3 and Table 2 are sufficient to understand the proposed system;
Answer: Thanks. Figure 2 and Table 1 have been eliminated.
6) The authors report that the microalgae sludge must be pressurized to 200 bar but in Table 4 the pressure is constant in all streams. Apparently, stream 3 should have a higher pressure;
Answer: Thanks. The text has been modified to be clearer.
7) The authors clarify how they obtained the equipment costs but do not present information on how they estimated the other costs (Table 7);
Answer: Thanks. Aspen Process Economic Analyzer software (APEA) allows estimating the costs of piping, civil, instrumentation, etc. once the costs of the equipment and the location of the plant are known. The text has been modified to clarify this aspect.
8) In economic analysis results, compare the optimal sale price obtained for the biofertilizer (3.5 €/kg) with the standard selling price of a chemical fertilizer.
Answer: Thanks for the comment. The price of 3.5 €/kg is not the optimal sale price, it has been considered as an adequate price for the economic analysis. The best sale price is the highest possible, which according to the characteristics of the biofertilizer and the values found in the literature could be in the order of 7.5 €/kg, but we decided to be conservative regarding this variable. The average price of simple nitrogen fertilizers in Spain in 2021 was below 0.5 €/kg, although currently due to inflation and the prices of gas and oil it must surely exceed that value. Biofertilizers have a much higher price because they have greater quality and higher production costs than traditional fertilizers and, therefore, are not comparable in price. To make a fair comparison of these two different types of products a life cycle analysis would have to be carried out in order to show the environmental impacts of each of these products and the costs involved in reverting negative environmental impacts caused by their use.